Fear and stressing in predator–prey ecology: considering the twin stressors of predators and people on mammals

http://orcid.org/0000-0002-8090-854X Fardell Loren L. 1 lfar3796@uni.sydney.edu.au
http://orcid.org/0000-0003-2162-8019 Pavey Chris R. 2
http://orcid.org/0000-0002-1067-3730 Dickman Christopher R. 1
1 School of Life and Environmental Sciences, University of Sydney , Sydney, NSW , Australia
2 Land and Water, CSIRO , Winnellie, NT , Australia
Lambert Max
Electronic publication date: 2020 Apr 30
Publication date: 2020
Volume: 8
Electronic Location ID: e9104
Received 2019 Sep 26; Accepted 2020 Apr 9
Copyright: © 2020 Fardell et al.
Copyright year: 2020
Copyright holder: Fardell et al.
License: This is an open access article distributed under the terms of the Creative Commons Attribution License, which permits unrestricted use, distribution, reproduction and adaptation in any medium and for any purpose provided that it is properly attributed. For attribution, the original author(s), title, publication source (PeerJ) and either DOI or URL of the article must be cited.
License URL: https://creativecommons.org/licenses/by/4.0/

Keywords: Fear, Stress, Anthropogenic, Urban, Endocrinology, Ethology, Landscape of fear, Cumulative, CEA, Predator

Funding: Holsworth Wildlife Research Endowment This work was supported by a Holsworth Wildlife Research Endowment. The funders had no role in study design, data collection and analysis, decision to publish, or preparation of the manuscript.

==============================
Predators induce stress in prey and can have beneficial effects in ecosystems, but can also have negative effects on biodiversity if they are overabundant or have been introduced. The growth of human populations is, at the same time, causing degradation of natural habitats and increasing interaction rates of humans with wildlife, such that conservation management routinely considers the effects of human disturbance as tantamount to or surpassing those of predators. The need to simultaneously manage both of these threats is particularly acute in urban areas that are, increasingly, being recognized as global hotspots of wildlife activity. Pressures from altered predator–prey interactions and human activity may each initiate fear responses in prey species above those that are triggered by natural stressors in ecosystems. If fear responses are experienced by prey at elevated levels, on top of responses to multiple environmental stressors, chronic stress impacts may occur. Despite common knowledge of the negative effects of stress, however, it is rare that stress management is considered in conservation, except in intensive ex situ situations such as in captive breeding facilities or zoos. We propose that mitigation of stress impacts on wildlife is crucial for preserving biodiversity, especially as the value of habitats within urban areas increases. As such, we highlight the need for future studies to consider fear and stress in predator–prey ecology to preserve both biodiversity and ecosystem functioning, especially in areas where human disturbance occurs. We suggest, in particular, that non-invasive in situ investigations of endocrinology and ethology be partnered in conservation planning with surveys of habitat resources to incorporate and reduce the effects of fear and stress on wildlife.

Introduction

Predators in many systems positively influence the local distribution and abundance of their prey (Polis et al., 1998; Ayal, 2007; Estes et al., 2011; Weissburg, Smee & Ferner, 2014), and indirectly but positively influence the functioning of whole ecosystems via trophic cascades (Prugh et al., 2009; Ritchie & Johnson, 2009; Estes et al., 2011; Ripple et al., 2014). Predators often elicit fear responses in prey that affect prey behaviour, energy budgets and the way they interact with the environment (Brown & Kotler, 2004; Clinchy et al., 2004; Romero, 2004). For example, the fear response of an individual of a prey species to a predator may be to change the locations or times in which they forage, which can have positive impacts on the structure of the environment (Laundré, Hernández & Ripple, 2010). Such fear responses can in turn influence ecosystem function (Schmitz, Krivan & Ovadia, 2004, Schmitz, Beckerman & O’Brien, 1997, Hawlena et al., 2012). These effects arise due to the ‘landscape of fear’ that prey individuals perceive—that is the spatial distribution of perceived predation risk that influences prey movement and behaviour as prey individuals attempt to mitigate risk and obtain essential resources (Laundré, Hernández & Altendorf, 2001; Laundré, Hernández & Ripple, 2010).

Recent studies suggest that the effects of predators are being altered, and often amplified, by human activities. For example, the demise of top predators in many parts of the world has ‘released’ populations of smaller predators, or mesopredators that have since become overabundant and now exert strong pressure on smaller species (Prugh et al., 2009). Predators that have been introduced to new areas also exert much stronger pressure on prey populations than do native predators (Salo et al., 2007). Climate change and associated shifts in primary productivity can also have bottom-up impacts on predator–prey interactions (Laws, 2017). Such novel alterations to predator–prey dynamics can depress prey populations and disrupt community structure, and as such require novel approaches to recognize and manage their effects (Carthey & Blumstein, 2018; Fleming & Bateman, 2018; Guiden et al., 2019).

Altered predator–prey interactions may contribute to declines in biodiversity, but a primary cause of biodiversity loss is the rapid global increase in human populations, driving agricultural, industrial, and urban expansions that change or destroy natural habitats (Madsen, Carroll & Moore Brands, 2010). Such expansions are resulting in novel human–wildlife interactions and secondary impacts that arise from human activities in close proximity to natural environments, and thus increasingly are becoming an additional stressor that influences ecosystem function. Pressures exerted both directly and indirectly by human activities have been likened to the pressures exerted by the presence of a top predator on prey (Frid & Dill, 2002; Rehnus, Wehrle & Palme, 2014; Patten & Burger, 2018). Natural predation pressures coupled with human-imposed predation-like pressures and/or additional exogenous stressors, such as pollution, arising from anthropogenic activities are likely to negatively affect prey species by increasing their levels or frequency of stress. Current research is developing a more nuanced understanding of the effects of human activity and predator presence on the stress of prey (Arlettaz et al., 2015; Jaatinen, Seltmann & Öst, 2014, respectively). However, few studies in terrestrial ecosystems have considered both of these pressures simultaneously or have linked behavioural responses with endocrinological evidence of stress. In one instructive recent study, human activity was shown to be positively correlated with physiological stress in ungulates, whilst in areas occupied by large predators stress was found to be lower and less variable (Zbyryt et al., 2017).

Conservation management and associated scientific research is often viewed from the perspective of a single discipline. However, ecological changes arising from anthropogenic disturbances, including the alteration of predator–prey encounter rates, creation of novel species interactions (including between people and wildlife), introduction of novel species, and the alteration of natural habitats, call for multidisciplinary solutions. This is particularly pertinent as urban edge habitats are becoming increasingly valuable global hotspots of wildlife activity (Ives et al., 2016), and human influences are spreading further into natural habitats (Otto, 2018). There is growing evidence that multiple introduced stressors (e.g. altered predator–prey interactions and disturbance from humans) can have compounding impacts on wildlife, especially when interacting with natural stressors in ecosystems (Geary et al., 2019). For these reasons, it is timely to review relevant information from the broad knowledge bases of conservation physiology, ethology, and ecology to communicate the urgent need for wildlife managers and urban planners to routinely consider how fear and stress effects from multiple sources, particularly novel predator interactions and human activities, can affect prey species and ecosystem functioning in increasingly altered systems. Our focus in this review is on mammals because of the wealth of information on this group, but we note relevant studies from other groups where appropriate. We begin our review by describing physiological responses of mammals to fear and stress, then we consider the behavioural responses of mammals to these factors, and argue that human activity should be considered as part of the ecosystem so that overall stress impacts can be managed accordingly. We then demonstrate how fear and stress can influence habitat use, how vegetation and microhabitat management potentially may be used to alleviate stress, and how to monitor for fear and stress to then create management change. Finally, we use Australia as an example to show the benefits of considering cumulative fear and stress impacts to mitigate the effects of introduced eutherian predators, increased human activity, and the possible interactive effects of the two.

Review Method

This is not a systematic review, but rather a novel synthesis of existing knowledge. It seeks to extend ideas on the physiological impacts of fear and stress, behavioural ecology, predator–prey dynamics, and conservation management to explore the influence of humans and altered predator–prey interactions in the urban environment. To ensure that all key topic areas were thoroughly covered we conducted literature searches via: Google Scholar©, Web of Science©, JSTOR© and Wiley Interscience Online Library©. Manuscripts were mined for information relevant to wildlife fear and stress in terms of ecological management under anthropogenic pressures, including introduced predators and altered predator–prey interactions. The literature searches were undertaken using the following key terms as part of their title, keywords, and/or within the abstract: ‘acute stress,’ ‘chronic stress,’ ‘homeostasis,’ ‘cumulative stress,’ ‘multiple stressors,’ ‘multiple threats,’ ‘allostatic load,’ ‘allostatic overload,’ ‘acclimitisation,’ ‘glucocorticoid response,’ ‘hypothalamic pituitary adrenal axis,’ ‘fear arousal,’ ‘fear evolution,’ ‘fear predation,’ ‘amygdala’ ‘fear,’ ‘Pavlovian fear conditioning,’ ‘glucocorticoid assay,’ ‘f(a)ecal glucocorticoid,’ ‘non-invasive glucocorticoid assay,’ ‘reactive scope model,’ ‘landscape(s) of fear,’ ‘risk allocation hypothesis,’ ‘olfaction’ + ‘fear’ + ‘mammal,’ ‘post-traumatic stress disorder,’ ‘predator’ + ‘odo(u)r’ + ‘fear,’ ‘predator cue(s),’ ‘predation stress hypothesis,’ ‘predator-sensitive food hypothesis,’ ‘human activity’ + ‘stress’ + ‘wildlife,’ ‘human disturbance,’ ‘interactive stress,’ ‘multiple stress(ors),’ ‘additive stress impacts,’ ‘synergistic stress impacts,’ ‘antagonistic stress impacts,’ ‘human’ + ‘wildlife’ + ‘resource subsidies,’ ‘predator trophic cascade,’ ‘wildlife urban adaptation,’ ‘urban ecology’ + ‘wildlife,’ ‘Australian mammal extinction,’ ‘critical weight range mammal,’ ‘diet’ + ‘change’ + ‘Australian’ + ‘predator,’ ‘red fox’ + ‘Australia,’ ‘domestic cat’ + ‘Australia,’ ‘human activity’ + ‘wildlife’ + ‘Australia,’ ‘Australia’ + ‘biodiversity’ + ‘conservation policy’ + ‘urban hotspot,’ ‘Australian environment protection and biodiversity conservation act’ + ‘cumulative stress,’ ‘Australia’ + ‘conservation’ + ‘glucocorticoids,’ ‘introduced predator control,’ ‘habitat structural complexity,’ ‘habitat structural diversity,’ ‘vegetation diversity,’ ‘vegetation heterogeneity,’ ‘habitat heterogeneity hypothesis,’ ‘cumulative effects assessments.’ Supplementary articles supporting the review narrative, suggested by reviewers on earlier versions of the manuscript, have also been included.

Mammalian Physiological Responses to Fear and Stress

Physiological stress can be broadly defined as a change in the physiological well-being of an individual following exposure to an aversive extrinsic stimulus—frequently referred to as a stressor (Selye, 1936). The physiological stress response that follows aims to restore internal homeostasis (Cannon, 1932). An individual animal experiences stress in response to conditions that threaten its survival or compromise its ability to maintain homeostasis. Examples include acute or chronic encounters with predators, inclement weather, significant natural disturbances including fire and flood, reduced oxygen availability and depleted food resources (Lima, 1998; King & Bradshaw, 2010; Malcolm et al., 2014; Santos et al., 2014; Crocker, Khudyakov & Champagne, 2016).

In mammals, activation of the hypothalamic-pituitary-adrenal (HPA) axis is a common stress response, but this may vary depending on the nature of the stressor (Mason, 1971); it is important to consider this when assessing stress impacts. Changes in abiotic conditions, such as in food availability, or the introduction of toxins or diseases can be stressful, but such changes do not arouse fear. However, a stress response involving peripheral autonomic and neuroendocrine changes (Yates, Russell & Maran, 1971; Sapolsky, Romero & Munck, 2000; McEwen & Wingfield, 2003), can be initiated by fear arousal (LeDoux, 2003; Labar & Ledoux, 2011). Animals with sophisticated nervous systems, such as mammals, exhibit a central motive state between threat stimulus and response that is driven by the amygdala (Pitkanen, 2000) and can be identified as ‘fear’ (Mineka, 1979; Öhman, 2000). Fear has been defined as a psychological state that triggers physiological responses to avoid or escape from a stressor (Epstein, 1972; LeDoux, 1996). The development of successful defense mechanisms to fear-inducing stressors has clear survival benefits for animals, and thus fear can be seen as a driver of evolutionary adaptations (Tooby & Cosmides, 1990). Common strategies of escape and avoidance allow animals to cope with recurrent stressors, such as the fear-inducing threat of predation (Lima & Dill, 1990).

The sections of the mammalian amygdala associated with fear behaviour serve as an interface between sensory input and information transport and processing, endocrine response, and motor output (Davis & Whalen, 2001). These interactions are associated with learning and memory via the involvement of the lateral and basal nuclei, as demonstrated on captive rodents using neurotoxic lesions on the basal and lateral nuclei of the amygdala (Wallace & Rosen, 2001), and are evident in Pavlovian fear conditioning paradigms (Davis, 1992; Maren, 2001). Activation of hormones, specifically glucocorticoids and norepinephrine in stress responses initiated by fear arousal provides feedback to the brain that influences emotion control and cognition, which contributes to fear conditioning (Rodrigues, LeDoux & Sapolsky, 2009). Fear responses may, therefore, be both conditioned as aversive learnt behaviours (Fanselow & Poulos, 2005), and unconditioned as innate freezing responses (e.g. Schulkin, Thompson & Rosen, 2003).

Given that fear motivates a stress response—initiating the freeze, and fight or flight actions—quantifying glucocorticoid outputs from the autonomic nervous and HPA systems should yield a measurable indication of fear from predation as a stressor. Minimally invasive techniques are available that assay glucocorticoid levels in faeces, fur, or feathers (Sheriff et al., 2011; Cook, 2012; Palme, 2019). Using minimally invasive methods to measure glucocorticoid levels in the wild does not necessarily yield an isolated ‘output’ of a fear or predator/human-induced stress response (Rosen & Schulkin, 2004). This situation results from glucocorticoids not being a molecule of fear per se but having a fundamental role in maintaining energy balance (Rosen & Schulkin, 2004). As such, glucocorticoids may change not just because of fear but also in response to other exogenous stressors, like food shortage (McEwen & Wingfield, 2003). Nevertheless, such methods could usefully compare longer-term stress responses among habitats or along disturbance gradients with variable exposure to predators, by following experimental designs that are considerate of additional stressors that may be present (MacDougall-Shackleton et al., 2019).

If two or more stressors are present, the resultant combined stress may present severe challenges to an individual’s physiological systems (Johnstone, Lill & Reina, 2012; Brearley et al., 2013; Malcolm et al., 2014; Arlettaz et al., 2015; Geary et al., 2019; Legge et al., 2019). Interactive fear and stress effects are context-dependent (Belarde & Railsback, 2016). They can be additive and combine the multiple impacts, synergistic whereby the presence of one threat amplifies another (Doherty et al., 2015), or antagonistic whereby one threat cancels the effects of the other. A reduction in the activity of mesopredators in the presence of humans provides an example of antagonism (Clinchy et al., 2016). Additive, synergistic, and antagonistic reactions have each been observed in mammalian prey in natural systems in response to exposure to multiple stressors (Crain, Kroeker & Halpern, 2008; Côté, Darling & Brown, 2016; Gunderson, Armstrong & Stillman, 2016; Jackson et al., 2016; Geary et al., 2019; Legge et al., 2019).

Exposure to a stressor(s) that is prolonged, constant, or recurring can have chronic impacts, as recovery from a stressor cannot occur whilst the threat remains (Sapolsky, Romero & Munck, 2000). Acute stress occurs as the initial response to a threat to sustain fitness in the short term; it subsides once the responding action—be it freezing, fighting, or fleeing—diminishes the threat (Wingfield & Kitaysky, 2002). Activation of the HPA axis in an acute stress response has rapid effects that increase immune system function, energise muscles via enhanced cardiovascular tone, and heighten cognition, including memory (Sapolsky, Romero & Munck, 2000). These responses occur via increased cerebral perfusion rates and use of glucose, all of which come at the cost of decreased appetite and reproductive behaviours (Sapolsky, Romero & Munck, 2000). Effectively, the acute stress response suspends non-essential behaviours in favour of altered behaviours that minimize the threat (Wingfield & Kitaysky, 2002).

Continued exposure to a stressor, or stressors, creates a state of chronic stress, which is classically described as allostatic overload (Dantzer et al., 2014). Allostatic load describes the body’s ability to maintain homeostasis in response to a stressor (Sterling & Eyer, 1988; McEwen & Stellar, 1993; McEwen & Wingfield, 2003). Allostatic overload, by extension, refers to the inability to maintain homeostasis and thus an organism’s increased susceptibility to external stressors (Sterling & Eyer, 1988; McEwen & Stellar, 1993; McEwen & Wingfield, 2003). Chronic stress reduces an organism’s resilience to future stressors by inducing extended behavioural changes in feeding, fighting, and mating (Mineur, Belzung & Crusio, 2006). Chronic stress also affects an organism’s physiological state by suppressing or impairing the reproductive system and decreasing physiological resistance to pathogens and toxins through the suppression of immune function (Dhabhar & McEwen, 1999; McEwen & Wingfield, 2003; Romero, 2004; Travers et al., 2010; Feng et al., 2012). Repeated exposure to a stressor can have population and community level consequences as homeostasis is compromised, and it increases susceptibility to additional stressors and can have flow-on effects. For example chronic risk of predation, which facilitates changes in nutrient demand for prey species and thus changes in the elemental composition of their excreta, can alter nutrient cycling in the community and ecosystem (Hawlena & Schmitz, 2010).

Cases of acclimatisation to chronic or repeated acute stressors have been observed, although the process often results in enhanced activation of the HPA axis to novel stressors, and thus may not be beneficial to fitness (Romero, 2004). Instead of acclimatising to a chronic or repeated stressor, glucocorticoid levels can remain the same, or become chronically elevated, or the HPA axis can shut down completely and render an animal vulnerable to future threats (Romero, 2004). Physiological impairment of the neurological, cardiovascular and musculoskeletal systems may also result from chronic stress: neurons of the brain can atrophy and impair memory, or grow and enhance fearfulness with extensive releases of adrenaline and cortisol (Roozendaal, 2000; McEwen, 2004); atherosclerotic plaques also may form and impede blood flow from repeated elevation of blood pressure (Manuck et al., 1988), and skeletal muscle can suffer severe protein loss (Wingfield & Kitaysky, 2002).

Although chronic and acute stress responses have been well defined for mammals in laboratory studies, results from in situ studies show that reactions vary with the stressor type that animals are exposed to, as well as habituation potential, food availability, social interactions, and density (Dickens & Romero, 2013). In situ mammalian endocrine responses to a consistent stressor, such as human activity, have not been studied sufficiently to be able to determine if there is a particular pattern of response. Avian endocrine responses to such stressors have been more extensively studied and they reveal that urban habitats, perceived as a consistent stressor, can shape endocrine responses in birds. However, a consistent stress response pattern has yet to be observed in birds (Bonier, 2012). Inconsistent behavioural responses of mammals and other animals to human activities that are often observed can perhaps be linked to modulating factors such as the level of human activity, the species and condition of the animal being observed, and the spatio-temporal context (Tablado & Jenni, 2017).

Mammalian Behavioural Responses to Fear and Stress: The Landscape of Fear Concept

The landscape of fear concept (Laundré, Hernández & Altendorf, 2001) postulates that prey are aware of microhabitat patches associated with high and low predation risk, where predators are either active and ubiquitous, or scarce (Laundré, Hernández & Altendorf, 2001; Shrader et al., 2008; Van Der Merwe & Brown, 2008; Laundré, Hernández & Ripple, 2010). Theoretically, landscape of fear effects should increase with landscape heterogeneity, since the differences between high and low risk sites become more pronounced and prey can more easily avoid high-risk sites (Bleicher, 2017; Gaynor et al., 2019). However, this relationship can depend on the species and condition of the habitat and, for some species, simple habitats can be the safest (Hammerschlag et al., 2015; Schmidt & Kuijper, 2015; Atuo & O’Connell, 2017). Landscape of fear effects will also be more pronounced in systems where interactions between predators and prey are less frequent (Schmitz, 2008), as is evident in findings from Pavlovian fear conditioning, and the ‘risk allocation hypothesis’—which states that animals exposed to constant high predation risk will increase their foraging risks over time (Lima & Bednekoff, 1999; Van Buskirk et al., 2002). Fear arousal in a landscape of fear results in two predictable outcomes: either avoidance of high risk areas, or modulation of behaviour (e.g. increased vigilance) to reduce predation risk when foraging in such areas (Gaynor et al., 2019). These outcomes indicate the potential positive benefits to conservation management of considering fear arousal and stress levels, and the direct and indirect cues that may trigger these effects (Atkins et al., 2017).

The likelihood of survival of a prey species is improved by having a well-developed perception of high predation risk, as it allows adaptive behavioural responses to be established (Bókony et al., 2009). Habitat shifts, temporal shift, grouping, vigilance and freeze, fly and fight responses are the most studied such responses to the non-consumptive effects of predators (Say-Sallaz et al., 2019). These often develop due to increased interaction rates between predators and prey. However, chronic stress impacts that are sufficient to affect reproduction and long-term survival can be experienced by prey species perceiving recurring predation risks (Thomson et al., 2010; Clinchy et al., 2011). Chronic perceived predation risk may also result in altered foraging activity driven by fear. This can affect where and what prey eat (Schmitz, Krivan & Ovadia, 2004), causing them to move from risky to sheltered microhabitats (Trussell, Ewanchuk & Matassa, 2006), in turn altering the distribution and availability of resources. Fear-based adaptive behavioural responses such as these underlie the landscape of fear concept.

For mammals, fear arousal can be triggered both by a predator’s presence or by a predator cue, such as an associated scent. Olfaction can be a key driver of fear arousal (Soso et al., 2014; Banks, Daly & Bytheway, 2016; Jones et al., 2016; Parsons et al., 2017). The mechanics of this are best understood in mammals: odours are detected by the accessory olfactory bulb that transmits information directly to the amygdala and hypothalamus, where fight or flight responses are developed (Fogaca et al., 2012; Canteras, Pavesi & Carobrez, 2015). Laboratory studies exploring the effects of post-traumatic stress disorder have exposed small mammals to predators or their cues to induce stress, and in doing so revealed that exposure to predator cues alone can affect the neural circuitry associated with fear (Rosen & Schulkin, 1998, 2004).

Subtle cues such as predator odours may precede threats and allow for a mammalian prey animal’s fear state to be conditioned to a cue that occurs before, or in correlation with, a previously encountered predation threat (Rescorla & Solomon, 1967; Rosen, 2004). Predation risk may, therefore, be perceived by prey species eavesdropping on predator scent marks, such as urine, faeces, or fur in the environment (Banks, Daly & Bytheway, 2016; Jones et al., 2016). Such odours have been observed experimentally to induce fear-like responses of freezing (Wallace & Rosen, 2000), vigilance (Nersesian, Banks & McArthur, 2012), fleeing (Anson & Dickman, 2013) and avoidance (Hayes, Nahrung & Wilson, 2006), across a wide range of species in both field and laboratory experiments (Apfelbach et al., 2005, 2015). Consequently, landscape of fear topography, where predators indirectly influence prey behaviour across a range of microhabitats, can arise from the influence of predator olfactory cues on mammalian prey foraging behaviours as much as it can from the direct threat of predation (Brown & Kotler, 2004; Parsons & Blumstein, 2010; Cremona, Crowther & Webb, 2014; Mella, Banks & McArthur, 2014; Hoffman, Sitvarin & Rypstra, 2016). It is worth noting, however, that whilst predator olfactory cues can elicit a fear or stress response, they do not always do so (Apfelbach et al., 2005). This may be because predators often leave the site after depositing a cue, and the cue intensity diminishes. Predator cues elicit responses that are modulated according to the differential intensity of the perceived threat, prior experience, or pending further information gathering, and thus can be perceived as a low risk by prey (Bedoya-Pérez et al., 2019).

Fear arousal due to predator presence or cues can deplete a prey individual’s energy budget, resulting in poor reproduction and health either via energy exhaustion from stress (i.e. the predation stress hypothesis: Boonstra et al., 1998; Clinchy et al., 2004; Romero, 2004; Støen et al., 2015), or reduced nutrition from foraging compromises (Brown & Kotler, 2004; Clinchy et al., 2016). As food resources become limited, additionally, prey may take greater risks to meet nutritional needs (i.e. the predation-sensitive food hypothesis: Sinclair & Arcese, 1995), but at a possible cost of additional stress. In areas of high human activity, this may result in wildlife increasing the stressors they are exposed to as they seek human-based food sources, such as waste or pet food, and in doing so increase their exposure to domestic pets, altered light, increased sound, and vehicles (Navara & Nelson, 2007; Morgan et al., 2009; Riley et al., 2014; Shannon et al., 2016; Doherty et al., 2017). For managers mitigating fear arousal, mapping landscapes of fear (Van Der Merwe & Brown, 2008; Kauffman, Brodie & Jules, 2010; Iribarren & Kotler, 2012; Smith et al., 2019) to identify, protect and extend safe foraging areas, could assist in the conservation of wildlife subject to human activity or multiple stressors such as human activity and predators.

Human Activity as a Fear-inducing Stressor

Human activity can influence wildlife in a wide variety of ways (Albert & Bowyer, 1991; Bowyer et al., 1999). Pressure from increased proximity to human activity or development that encroaches upon animal home ranges results in wildlife becoming either: (1) ‘urban avoiders’ that move away from human activity; (2) ‘urban adapters’ that make some use of anthropogenic resources but still rely largely upon those found naturally; or (3) ‘urban exploiters’ that are synanthropic and make full use of anthropogenic resources (McKinney, 2006). The effects of human activity can therefore differ between species and trophic levels, and even within the same species, depending on personality. Furthermore, the negative effects of a disturbance on one species may result in flow-on responses of positive consequences for its prey, predators, or competing species (Gill, Sutherland & Watkinson, 1996; Crooks & Soulé, 1999; Leighton, Horrocks & Kramer, 2010). These effects, however, depend strongly on the availability of necessary resources within the human occupied/disturbed areas, and will often differ between systems.

It has been postulated that humans may impose widespread effects on ecosystem function if they induce greater fear responses in ubiquitous small predators than in top predators (Clinchy et al., 2016), or by increasing physiological stress in large ungulates through antagonistic behaviours (Vijayakrishnan et al., 2018). Indeed, human activity can be comparable to the above-mentioned impacts of predation, creating landscapes of fear (Hofer & East, 1998; Frid & Dill, 2002; Rehnus, Wehrle & Palme, 2014; Patten & Burger, 2018), and disrupting foraging (Ciuti et al., 2012; Clinchy et al., 2016). Humans can also act as ‘super-predators’ that disproportionately kill carnivores and drive trophic cascades (Darimont et al., 2015). Trophic cascades can also arise when human activity has non-consumptive effects on the ecological roles of large predators (Smith et al., 2017). For example large predators may increase their hunting efforts in urban areas: if their own fear response to human activity results in less time spent consuming a kill, this in turn may increase their kill rate of prey in urban areas to meet their energy needs (Smith et al., 2017). When large predators avoid human activity, and/or meso-predators reduce their foraging around human activity, prey species may respond by increasing their own foraging activity (Suraci et al., 2019), effectively taking refuge behind the ‘human shield’ that is reducing predation pressure for them (Berger, 2007; Leighton, Horrocks & Kramer, 2010; Kuijper et al., 2015).

Human activity can affect the trophic structure of communities not only via top-down pressure, but also by exerting bottom-up pressure through the constant provision of alternative resources (Fischer et al., 2012), such as subsidies of shelter, water and food with year-round primary production. These resources can outweigh the stresses of human disturbance and influence population and community behaviours (Parris, 2016), resulting in urban colonisation by wildlife (Shochat, Lerman & Fernández-Juricic, 2010; Jokimäki et al., 2011). As such, human activity does not always result in fear responses, particularly for prey species that benefit from the human shield and access to food, water or shelter subsidies (Lyons et al., 2017). It is possible that, given these conditions, and provided that resource subsidies are nutritionally rich and accessing them does not increase pathogen transmission risk (Murray et al., 2016), prey species like small mammals would experience fitness benefits in urban areas. Resource subsidies can also be exploited by predators, resulting in increased predator activity in urban areas but also in altered foraging behaviours that either decrease or increase predation on small prey (Iossa et al., 2010; Bateman & Fleming, 2012; Fischer et al., 2012; Newsome et al., 2014, 2015).

Despite such positive effects of human activity on wildlife, negative impacts are more pervasive for many species. The negative effects of high levels of disturbance, for example from roads and vehicles, or of active interference from recreational activities conducted in designated conservation areas, can lead to physiological stress or displacement, countervailing any positive effects of humans on wildlife (Kloppers, St. Clair & Hurd, 2005; Banks & Bryant, 2007; Berger, 2007; Ciuti et al., 2012; Rehnus, Wehrle & Palme, 2014; Arlettaz et al., 2015; Patten & Burger, 2018; Vijayakrishnan et al., 2018).

A common response in mammals to increased exposure to human activity is a temporal shift in activity patterns, from diurnal to crepuscular or nocturnal, to avoid interaction with humans (McClennen et al., 2001; Tigas, Van Vuren & Sauvajot, 2002; Riley et al., 2003; Ditchkoff, Saalfeld & Gibson, 2006; Gaynor et al., 2018); the same or reverse may occur too, to reduce interactions with predators (Brown, 2000; Laundré, Hernández & Altendorf, 2001; Kohl et al., 2018; Smith et al., 2019). Temporal shifts can have strongly negative effects if they limit a forager’s ability to locate and capture prey (Ditchkoff, Saalfeld & Gibson, 2006). Regardless of the response, it is evident that human activity can have profound indirect effects on community interactions through altering individual behaviour, particularly foraging, through either fear arousal or a stress response (Frid & Dill, 2002; Werner & Peacor, 2003).

The ability of wildlife to cope with urban environments can occur through either plastic or evolved shifts in behaviour, foraging, food preferences, or predator avoidance, and in shifts in the timing of breeding (Ditchkoff, Saalfeld & Gibson, 2006; Møller, 2009; Shochat, Lerman & Fernández-Juricic, 2010; Rodriguez, Hausberger & Clergeau, 2010; Alberti, 2015; McDonnell & Hahs, 2015; Otto, 2018; Santini et al., 2019). It is most common for urban-occupying species to plastically adjust to human imposed stressors (Donihue & Lambert, 2015), with some ultimately evolving tolerance to urban environments. Human activity can therefore impose pressures that are strong enough to influence changes in behaviour and in physiology—endocrinology in particular (Bonier, 2012; Snell-Rood & Wick, 2013; Otto, 2018). Bolder individuals are most likely to exploit novel opportunities and have reduced stress responses to human activity (Atwell et al., 2012). Animals that routinely encounter human activity, typically provide parental care, and are capable of reproducing multiple times across their life span may produce offspring that are acclimated to human activity and accordingly show lower fear/stress responses (Schell et al., 2018). Habituation of groups of animals to human activity is being increasingly observed, often as a result of bold individuals spending significant amounts of time around human-disturbed areas and displaying reduced fear responses (Stillfried et al., 2017). This counters the idea that human activity always imposes landscapes of fear, but may reflect the combination of reduced resource availability and bold personalities that allow for urban habituation (Lowry, Lill & Wong, 2013). In some instances this may be problematic for humans if increased interactions with habituated animals increase vehicle accidents and/or transfer of zoonotic diseases; this highlights the potential need to manage some mammalian prey species in urban areas (Honda et al., 2018).

Fear and acute/chronic stress may be constant hurdles faced by wildlife, but adding the effects of anthropogenic disturbance could result in the elevation of acute stress responses to prolonged and widespread chronic stress, and upscale the possible impacts from individuals to populations (Rehnus, Wehrle & Palme, 2014). As we have outlined, human activity can both indirectly and directly create landscapes of fear, influencing and amplifying existing landscapes of fear through interactive effects. Predator presence and human activity can also interact to alter ecosystem structure and modify the predation risk perceived by prey species, with either positive or negative impacts dependent upon the circumstances. The combination of stress resulting from the pressures of altered predator–prey interactions and human activities is yet to be investigated extensively. Understanding the cumulative effects that both may impose is critical for effective conservation of wildlife in an increasingly human-influenced world.

Alleviating Fear and Stress for Wildlife Conservation

As we have illustrated, fear can produce stress and simultaneous multiple stressors can have chronic effects that negatively influence both predator and prey species populations (Allan et al., 2013). To synthesize these impacts, we note that in modern urban situations both habitat stressors and introduced stressors now collide (Fig. 1). Habitat stressors result from naturally occurring environmental factors such as native predator presence (especially for prey), social and reproductive pressures, limited access to refuges/nests and/or food and water, and disease and parasite prevalence, all of which produce predictive homeostasis responses by prey species (Fig. 1). Introduced stressors, by contrast, are additional stressors that are imposed by human actions that result in reactive homeostasis responses. These introduced anthropogenic stressors include structural developments that change landscapes, human recreational activities in natural habitats (including use of vehicles), introduced pollutants and toxins that alter landscapes and resource quality, and introduced predators and altered predator–prey interactions (Fig. 1). The combination of impacts from both naturally occurring and introduced anthropogenic stressors has the potential to cause a cumulative stress response that can result in homeostatic overload or failure, as defined by Romero, Dickens & Cyr (2009), and may result in population collapses due to increased susceptibility of individuals to the additional stressors (as we have demonstrated above, and Fig. 1). Cumulative stress may be additive or synergistic, and is likely to be particularly detrimental for prey species. For example, if prey species face multiple stressors they may take greater foraging risks, or be less able to allocate energy to vigilance or flight behaviours, and thus become more susceptible to predation or additional stressors. In the case of population-impacting stressors, the local population may be impacted to a high degree such that it becomes threatened (Sweitzer, 1996; Doherty et al., 2015). As such, conservation action in areas where simultaneous introduced stressors occur in addition to natural ecosystem pressures, like in urban and peri-urban habitats, may be urgently needed.

Figure 1 A Venn diagram showing two main categories of stressor.

On the left are stressors that occur naturally in ecosystems, such as native predators, social and breeding interactions, availability of refuge and burrow/den microhabitats, disease/parasite prevalence, and availability of food and water. On the right are introduced stressors, primarily those arising from anthropogenic disturbances, such as urban developments, human activity, introduced toxins and introduced predators. Where both categories of stressor occur together simultaneously, as in many urban environments, cumulative stress impacts can result in homeostatic overload or failure (as defined by Romero, Dickens & Cyr (2009)). In these situations populations may be at particular risk of collapse and conservation action will be most urgent.

The most direct way to alleviate stress in wildlife is to remove or reduce the key stressors. This may be difficult if the stressors are anthropogenic, as the needs of the expanding human population often usurp those of wildlife; alternative solutions then need to be considered. A suite of activities related to managing vegetation structure, density, and/or heterogeneity may well provide such an alternative solution. These activities result in increased availability of potential refuge sites and may be readily achieved through ongoing habitat engineering that, for example supports the growth of structurally complex plants or adds logs/rock piles to reduce areas of open space. In order to mitigate introduced fear and stress effects, it is important that future studies investigate whether vegetation management and other habitat components can alleviate some of the pressures associated with multiple introduced stressors for target species.

Managing Habitat to Alleviate Wildlife Fear and Stress

The impact of predators (or human activity) can be mitigated by changing the configuration of risky areas within a habitat (Hopcraft, Sinclair & Packer, 2005; Lone et al., 2014). Food availability often drives habitat selection (Sherman, 1984; Johnson & Sherry, 2001), but predation risk and human activity affect a suite of correlated factors such as movement decisions (Turcotte & Desrochers, 2003), foraging patterns (Gil, Zill & Ponciano, 2017), social organisation (Rodríguez, Andrén & Jansson, 2001) and reproductive success (Zanette et al., 2011). Supplementation of essential resources including refuges, nests or roosts, food and water to wildlife is a growing conservation method due to its ease of implementation, immediate results, and favourable portrayal in the media. One example of positive results includes increased parasite resistance in nestling birds after supplementation with high-quality food during the stressful young-rearing stage (Knutie, 2020). Another example is the reduction in impact from introduced predators on small desert mammals following the addition of artificial refuge structures (Bleicher & Dickman, 2020). Despite these successes, negative impacts can also ensue following supplementation. Built-up refuges may be primarily investigated by invasive species, such as the black rat (Rattus rattus), which could potentially deter commensal native species and support the population growth of invasive species (Price & Banks, 2018). Artificial nests or roosts can support increased predation rates owing to the structure supporting a sit-and-wait predator at the exit (McComb et al., 2019). Food and water subsidies can increase pathogen transmission, reduce movement and migratory behaviours, increase predator–prey interaction rates, and competitor aggression interactions (Murray et al., 2016). However, owing to the benefits of supplementation, especially during times of stress, it may be used as a tool to mitigate the negative impacts of stress on wildlife if conducted in a targeted, monitored and well-considered manner (Freeman et al., 2020).

Considering the existing knowledge base on food and water supplementation, meta-analyses have suggested that the associated disease risk may be managed by focusing on specific natural food sources and pathogen groups subject to the target ecosystem and by increasing the spatial extent of feeding stations across potential linked habitats, which also limits microparasite host aggregations (Becker, Streicker & Altizer, 2015; Becker et al., 2018). In considering the specific needs of target animals and ecosystems, chainsaw-carved cavities have more favourable thermodynamic properties for small mammals and birds than do artificial next boxes (Griffiths et al., 2018). As they mimic natural hollows they are also less likely to increase rates of predation. Supplementation projects are therefore moving forward using this growing knowledge base to adaptively tailor methods to specific conditions, and to orchestrate pre- and post-monitoring that ensures successful stress mitigation (Civitello et al., 2018).

Survivorship of prey, in the face of predation stressors, is often positively correlated with increased structural complexity of the habitat (Hopcraft, Sinclair & Packer, 2005; Lone et al., 2014; Leahy et al., 2016). This has been demonstrated in many studies to be a consequence of the increased opportunities for prey to escape and hide, thereby mediating predator–prey dynamics by reducing encounter rates, as only a proportion of the total prey population remains available to predators (Holt, 1984; Kotler & Brown, 1988; Bianchi, Schellhorn & Van der Werf, 2009; Rieucau, Vickery & Doucet, 2009; Klecka & Boukal, 2014; Laundré et al., 2014). The extensive research on the benefits to prey of habitat structural complexity builds on the ‘habitat heterogeneity hypothesis,’ which postulates that structurally complex habitats support increased species diversity by offering a wide range of niches and diverse ways of exploiting resources (Bazzaz, 1975). Habitat complexity, by extension, has been generalized to be a primary driver of biodiversity (Pianka, 2011). The landscape of fear concept accepts this principle too, in that a wide range of microhabitat types offers multiple foraging and shelter conditions with varied predation risks for species. For example northern quolls (Dasyurus hallucatus) of the semi-arid Pilbara region in Western Australia utilise complex rocky habitats in preference to open grasslands where the threat of predation from feral cats (Felis catus) is greater (Hernandez-Santin, Goldizen & Fisher, 2016).

Animals in habitats with high predation pressure may display foraging preferences for microhabitats or times that they perceive to be safe (Brown, 2000; Laundré, Hernández & Altendorf, 2001; Laundré, Hernández & Ripple, 2010). Some small mammals seek structurally complex vegetation owing to the reduced risk of predation and increased reward of foraging they find there (Lima & Dill, 1990; Andruskiw et al., 2008). Others, such as the Australian hopping mouse (Notomys alexis) that exhibit bursts of speed in open habitat, are less adept at moving through complex vegetation (Spencer, Crowther & Dickman, 2014). No matter the species, in landscapes of fear a prey species’ use of the topography, refuges, and escape substrates can indicate its perceived risk of predation (Brown & Kotler, 2004; Van Der Merwe & Brown, 2008; Shrader et al., 2008), and the associated fear and stress that may arise or be alleviated due to habitat structure. The same principle should be relevant to reducing stress in proximity to human activity, considering that prey responses to human activity and predators are similar.

Human activity and disturbance can quickly reduce habitat complexity (Western, 2001). However, enhancing habitat complexity and heterogeneity is being incorporated increasingly into restoration and management efforts, with some success (Brown, 2003; Bernhardt & Palmer, 2007; Palmer, Menninger & Bernhardt, 2010). To balance the needs of people and biodiversity, local planning procedures are at times now incorporating green spaces and greening initiatives into urban areas. As habitat complexity and diversity are of particular importance in supporting biodiversity and population sustainability, it is important that habitat structure underpins the engineering of wildlife habitats in urban and urban-adjacent areas (Threlfall et al., 2016). The effectiveness of any management regime depends on recognising the direct and indirect impacts that occur across sustainable ecosystems. Consequently, habitat complexity as a management objective requires that urban landscapes are approached on a case by case basis, with full assessment before the habitat is ecologically engineered (Tews et al., 2004).

Management Tools to Observe and Alleviate Fear and Stress for Wildlife Conservation

Cumulative stress maps of human activity and stressors that occur naturally in ecosystems have allowed returns on restoration investments to be maximized by indicating key areas that will or do need intervention to mitigate the effects of severe stress on wildlife, as opposed to unsuccessful piecemeal management that focuses on one or two stressors broadly (Allan et al., 2013). Physiological and behavioural stress in wildlife may be cost-effectively and straightforwardly observed by comparing the outcomes of simultaneous measurements taken across a disturbance gradient where multiple stressors are both abundant and low, using several well-established methods. Specifically, assays of faecal/urine/fur/feather glucocorticoid metabolites can provide insight into the level of physiological stress an animal is experiencing under comparative circumstances (Cook, 2012; Cooke et al., 2013; Sheriff et al., 2011; Palme, 2019). Pairing such results for each location with those of giving-up density surveys (Brown, 1988) that can also be filmed by infrared motion sensor cameras to observe foraging behavioural responses (Leo, Reading & Letnic, 2015), yields an understanding of both physiological stress responses and foraging responses. Comparing these results may further indicate if behavioural responses are changing to moderate physiological stress impacts (Carlstead, Brown & Seidensticker, 1993; Carlstead, Brown & Strawn, 1993). Simultaneous measurements of habitat quality that include vegetation cover, refuges, distance to food/water source, and distance to disturbances (e.g. human activity or predator sighting based on spotlighting or in-field camera traps) will build a dataset for spatial correlation analyses that can geographically map the landscape of fear. Such measurements should also identify what factors influence the landscape, such as by use of behavioural data (Willems & Hill, 2009) and/or giving-up density data (Van Der Merwe & Brown, 2008).

Spatial mapping of the physiological stress response, based on the collection locations of samples, could be overlaid and compared to these results also. Such spatially mapped results may clarify important areas that are in need of protection or the habitat types that could be extended to mitigate stress impacts. It is worth noting that several authors have emphasized that these methods need to be used correctly to ensure that stress and habitat factors are linked (e.g. McMahon et al., 2018; Bleicher, 2017; Bedoya-Perez et al., 2013; MacDougall-Shackleton et al., 2019). For example collection of fresh faecal samples of a target species along a gradient of high to low human activity would allow glucocorticoid metabolites to be assayed, providing insight into potential stress induced by human activity along the gradient (Rehnus, Wehrle & Palme, 2014). A similar approach could be used to assay stress perceived by prey species in areas of high and low predator activity, and sex hormones also could be assayed to explore sex-related effects of stress. By allowing pathways of fear and stress to be mapped, including sites where stressors from predators and human activity occur simultaneously, such methods should allow managers to reliably identify where best to intervene to preserve at-risk populations and maintain community stability. We consider potential interventions in more detail below.

If cumulative stressors occur and are suspected to have detrimental effects on target wildlife populations or communities, we suggest that a potentially powerful mitigation approach could be developed based on Cumulative Effects Assessments (CEA). Devised in the 1990s amid growing concerns that Environmental Impact Assessments (EIA) did not consider all the effects of urban and peri-urban developments (Smit & Spaling, 1995), CEAs have been advocated as an effective tool for use by on-ground practitioners (Duinker et al., 2013). Numerous countries have mandated that CEAs be incorporated into EIAs, namely: the United States of America, Canada, Europe, New Zealand, and Australia each advocate some level of CEAs (Therivel & Ross, 2007). Despite this legal requirement, and the CEA concept being widely known in scientific literature, it is rarely applied in practice (Ma, Becker & Kilgore, 2009; Foley et al., 2017).

We propose further that the principles of the ‘reactive scope model’ be used to develop CEAs, to identify where cumulative stressors occur, and thus better inform conservation management initiatives in areas where wildlife is subject to homeostasis overload or failure (Fig. 2). The results from the above mentioned methods and/or other applicable established methods may be used to answer our suggested CEA questions. The reactive scope model (Romero, Dickens & Cyr, 2009) provides useful insight into the range of physiological mediators available in response to a stressor. It maps the homeostasis range of a given species in four stages: (1) predictive—change occurs in response to routine environmental change, such as seasons or day to night; (2) reactive—change occurs in response to an unpredictable change, allowing survival via classic stress responses; (3) overload—consistent changes occur in response to a stressor, and chronic stress impacts start to occur; and (4) failure—the species shows inability to sustain homeostasis, and is very susceptible to additional stressors and death (Romero, Dickens & Cyr, 2009). If our proposed CEA approach (Fig. 2), based on the reactive scope model, were applied to small mammals in urban, urban adjacent, and peri-urban ecosystems, for example then areas of conservation concern could be identified where the additive or synergistic impacts of human disturbance and introduced predators combine with stressors that occur naturally. Appropriate management, such as increasing habitat complexity or reducing human activity in conservation areas (Bleicher & Dickman, 2020), could then be implemented to alleviate these stressors.

Figure 2 A conservation management approach that outlines the key steps for assessing if cumulative stress impacts are occurring between stressors that occur naturally in ecosystems and those that are introduced.

The circled areas indicate where conservation management initiatives may be used to mitigate these effects through management of vegetation complexity or supplementation materials such as water stations or carved ‘tree-hollow’ cavities.

Fear and Stressing in Small Mammal Ecology in Australia; The Need to Consider the Twin Stressors of Introduced Predators and People in Wildlife Management

Small mammals are often somewhat resilient to threatening processes owing to their high population growth rates (Cardillo et al., 2005, 2006). However, in Australia small mammal populations are declining quickly, and due to causes dissimilar to those driving global declines (Woinarski, Burbidge & Harrison, 2015). Declines in Australia have been attributed to a wide range of habitat stressors: habitat loss, altered fire regimes, disease, increasing temperatures, decreasing water availability, depleted soil quality and salinity (Woinarski, Burbidge & Harrison, 2015). However, mammals within a ‘critical weight range’ (CWR: 35–5,550 g) are particularly vulnerable (Chisholm & Taylor, 2007; Woinarski, Burbidge & Harrison, 2015) owing to the effects of predation by two introduced carnivores, the red fox (Vulpes vulpes) and domestic house cat (Felis catus) that arrived in Australia over 150 years ago (Johnson, 2006). If predators are introduced, their impacts on prey are likely to be exacerbated owing to prey naïveté (Doherty et al., 2016). Such impacts have been particularly acute on wildlife in Australia, where eutherian carnivores are recent arrivals (Salo et al., 2007). However, the question of Australian native animal stress responses and the extent of their naiveté to these introduced eutherian predators is debatable (Banks & Dickman, 2007; Carthey & Banks, 2016). Although CWR mammals are of high conservation concern, predation from introduced predators poses a threat also to all native predators by reducing their food resources, which in turn may increase predation and predation-associated stress on alternative food sources such as smaller (<35 g) or larger (>5,550 g) mammal species. For example a study examining the diet of a nocturnal, avian predator, the sooty owl (Tyto tenebricosa) before and after red fox introduction, revealed a dietary shift post introduction, with owls consuming more arboreal than terrestrial prey species after fox arrival (Bilney, Cooke & White, 2006). This shift to consuming arboreal prey increased dietary overlap with the sympatric powerful owl (Ninox strenua), providing disproportionate predation pressure on prey in the ecosystems of East Gippsland, Victoria.

Predation by red foxes and cats is prevalent not only in natural habitats but also in agricultural and urban habitats (Dickman, 1996; Morgan et al., 2009; Bino et al., 2010). The paths and roads that fragment urban and agricultural habitats are used frequently by these predators, which exacerbates their predation pressure on prey species by combining with impacts imposed by human activities (Latham et al., 2011, Červinka et al., 2013). Consequently, this can result in increased abundances of red foxes and cats in human-modified landscapes that border or include natural habitats (Towerton et al., 2011; Graham, Maron & McAlpine, 2012). Urban and agricultural habitats present many threats to wildlife, but they may also offer food and shelter opportunities (Pickett et al., 2001; Gaston et al., 2005; Hobbs, Higgs & Harris, 2009). As reported by Ives et al. (2016), 46 percent of threatened Australian animals occur in or near Australian cities. Thus, the fate of many species could depend on accommodating their needs in urban and agricultural habitats (Ives et al., 2016). A recent assessment of data collected at the Wildlife Rehabilitation Centre in Queensland Zoo, Australia, revealed that pet cat or dog attack, car strike, and entanglement in human-placed objects represented 56.4% of the causes of submission of injured wildlife; mortality rates associated with these traumas were also high, with 61.3% of admitted animals dying from their injuries (Taylor-Brown et al., 2019). These threats may contribute to landscapes of fear, through fear arousal, altered foraging behaviour, post-traumatic stress reactions, and cumulative stress exposures resulting in chronic stress responses.

Despite growing knowledge of anthropogenic impacts on threatened species, Australian conservation planning continues to exclude urban and agricultural habitats from consideration (Dales, 2011). There is recognition of the scope of this issue in Australia (Hill, Carbery & Deane, 2007; Carthey & Banks, 2012; Threlfall, Law & Banks, 2012; McCauley, Jenkins & Quintana-Ascencio, 2013; Banks & Smith, 2015; Ives et al., 2016), but few studies have explored the interactive effects of introduced predators and human activity on the survival of prey species. Despite the development of glucocorticoid analysis techniques to determine stress in animals dating to the 1960s (Jones et al., 1964), only six of the 60 extant small mammals of conservation concern in Australia have been subject to studies seeking to better understand their glucocorticoid response to stressors (Hing et al., 2014), such as predation by introduced species or human activity.

There is a need for alternative management regimes to mitigate the additional stressors of human-imposed impacts on wildlife, and the situation we have outlined in Australia underlines the urgency of this need. As there are difficulties with the current control of introduced predators in Australia (Glen & Dickman, 2005; Rayner et al., 2007; Bergstrom et al., 2009; Norton, 2009; Warburton & Norton, 2009; Carroll, 2011; Newsome et al., 2017), there is also an urgent need for alternative management regimes to mitigate the additional stressors of introduced predator-imposed impacts. As we have argued throughout this review, it is logical that biodiversity conservation managers consider stress from the combined impacts of introduced predators, altered predator interactions, human activity, and stressors that occur naturally in ecosystems. Using the example methods to conduct our proposed CEA, based on the reactive scope model, to draft appropriate management plans, such as increasing habitat complexity or reducing human interaction in areas, would be a progressive response towards preserving many species of Australian small mammals and their constituent communities.

Conclusion

Considering ongoing global urbanization and the acknowledged importance of urban areas to biodiversity conservation, there is a great need for increased focus on the management of urban biodiversity. Management decisions require information about fear and stress impacts on wildlife, including impacts from both human activities and predators, especially if they are novel or introduced. Understanding the impacts of human activities is a research priority for modern science. There are many gaps in our current understanding of fear and stress impacts on wildlife, and of associated impacts of altered predation pressures and the persistence of target populations. To sustain biodiversity in urban and urban-adjacent green space habitats, reserves and conservation areas, it is vital that we establish better understanding and management of the multiple stressors that operate in these systems.

Additional Information and Declarations

Competing Interests

Author Contributions

Data Availability

The authors declare that they have no competing interests.

Loren L. Fardell conceived and designed the experiments, performed the experiments, analyzed the data, prepared figures and/or tables, authored or reviewed drafts of the paper, and approved the final draft.

Chris R. Pavey conceived and designed the experiments, authored or reviewed drafts of the paper, and approved the final draft.

Christopher R. Dickman conceived and designed the experiments, authored or reviewed drafts of the paper, and approved the final draft.

The following information was supplied regarding data availability:

This is a literature review article and did not generate raw data.

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
