# Peer review of "Fear and stressing in predator–prey ecology: considering the twin stressors of predators and people on mammals"

_PeerJ, doi:10.7717/peerj.9104_

## Round 0.1 · original submission · Major Revisions

Three reviewers and myself have now assessed your manuscript. All reviewers believe that the goals set out for this review do indeed address a hole in the literature. All reviewers are keen on the topic of this manuscript but feel in many ways that the organization of the review was a bit too jumbled to accomplish the goal of a cohesive and synthetic review that brings together several fields of research. That being said, all reviewers have offered substantial feedback for improving the clarity and organization of the manuscript and have identified numerous additional citations that should help fill some gaps in this review.

The reviewers all provide substantial useful feedback but I highlight that all three reviewers identify the need for operational definitions that are clearer and placed immediately upon first use. I agree with this sentiment, particularly when trying to merge several fields of literature.

One topic that additionally caught my eye was "synurbanization" or "adapatation" of wildlife on line 315. I've never heard of this term and am not sure these are the best citations for your argument. My colleagues and I have been writing about whether species rapidly evolve (i.e., adapt) or plastically adjust (i.e., acclimate) to urbanization. See the Donihue and Lambert or Alberti citations below as well as a number of empirical studies that have since come out after these perspectives. I think parsing out whether predator and prey are evolving or acclimating is actually quite interesting, particularly with this idea of stress. For instance, Schell et al recently found that coyote parents increasingly acclimated to human disturbance over time, passing on this plastic acclimation to later pup cohorts. I think thinking about these responses and possible mismatches between rates of acclimation or adaptation between predators and prey may be useful here.

Donihue and Lambert: https://link.springer.com/article/10.1007%2Fs13280-014-0547-2

Alberti: https://www.cell.com/trends/ecology-evolution/fulltext/S0169-5347(14)00249-3

Schell et al: https://onlinelibrary.wiley.com/doi/full/10.1002/ece3.4741

Additionally, this review seems heavily mammal-focused but not explicitly so. Even one of your search terms includes mammals. I would make sure to be explicit about whether this review broadly encompasses all wildlife with a shared stress response or is mammal-oriented. Reviewer 3 makes notes of stress variation in birds and I note that the stress response in reptiles and amphibians can be quite heterogeneous with respect to predators and anthropogenic stressors.

I look forward to a revised version of your manuscript.

·

Basic reporting

Fardell et al. set up a nice review topic that is both cross-disciplinary (bridging ecology/conservation and neuroscience/endocrinology) and novel (in my experience). My main concern throughout the manuscript is that while the authors set up this exciting idea of using fear/stress to inform conservation, the main text of the article does not accomplish this stated goal. Without carrying this central theme throughout the manuscript, the authors risk writing another general "predator-prey in novel habitats" review, of which there are many (e.g., in the last two years alone: Carthey and Blumstein 2017 Trends in Ecol Evol, Fleming and Bateman 2018 Animal Behavior, Gaynor et al. 2019 Trends in Ecol Evol, Guiden et al. 2019 Trends in Ecol Evol). However, I would personally like to see a revised version of the manuscript, because I am genuinely interested to see if routine endocrinology measurements could be used to better conserve animals in highly novel habitats.

Along these same lines, many conservation professionals likely think of “stress” and “fear” as nebulous abstract concepts. My perspective as a behavioral ecologists is that they are definitely important, but I think this paper would benefit from an explicit operational definition of stress, fear, etc. early in the introduction. This would then prime the reader for the sections describing potential applications of this work (Lines 420-516). I think the authors are really on to something important here, but to accomplish this we need something more concrete to frame our thinking.

As written, the “hook” of this paper is all about the combined stress effects of human activity and introduced predators. However, I feel like much of the writing does not focus on introduced predators, but rather could describe any predator at all (as suggested at Line 60). I’m also not entirely sure why the authors suggest that introduced predators will be impose more stress on their prey than native predators (Line 63), as prey often do not even recognize introduced predators as dangerous (Kuehne and Olden 2012 Freshwater Biology). Carthey and Blumstein (2017) Trends in Ecol Evol is a highly relevant paper here that I think should be cited. I suggest that either a) the authors rephrase the hook to more broadly describe predation (from native or non-native species) in conjunction with human activity, or b) try to more explicitly link previous studies specifically focusing on introduced predators throughout the manuscript. Do we really need introduced predators to get the main point here?

Experimental design

Line 99-127: Since this isn’t a systematic review or meta-analysis, I’m not sure that this section brings anything to the table. Perhaps this is a journal requirement, but I personally see no reason for this to be included in the main text.

Throughout the paper, I could quibble with some of the citations, but I think there are a number of good reviews on related (but more general) topics (see my comment in Section 1: Basic Reporting). On line 301 though, I strongly recommend citing Gaynor et al. (2018) Science--a highly relevant study here.

Validity of the findings

As a conservation biologist, I really enjoyed the authors’ section describing the different stress mechanisms experienced by animals. However, I found this level of depth missing from much of the manuscript (e.g., Lines 267-339, Lines 341-392). Based on the Introduction, I expected the majority of this manuscript to focus on how human activity and introduced predators alter the endocrinology, etc. of their prey, yet this critical detail is absent from most of the paper. If the authors are correct in arguing that successful conservation will account for these physiological stressors, at the bare minimum they need to include speculative hypotheses describing how humans or predators are affecting prey endocrinology, but hopefully cite more relevant papers here that have already started to look into this.

I think it’s great that the authors are thinking about how their research could be applied to conservation (Line 475-516), and again I think this paper could stimulate some really exciting research. However, in order to be applicable, I think this section needs more detail and guidance. Once fecal samples are collected, how should we interpret the results? What are some caveats to fecal glucocorticoid sampling? Similarly, the authors describe the reactive scope model but don’t actually attempt to apply this to a CEA. Before this could be implemented, it needs to be much more thoroughly explored.

Additional comments

In addition to the big-picture comments below, I have a number of minor line-by-line comments.

Line 27: I wonder if this statement is biased towards conservation in Australia, where introduced predators represent an especially urgent threat toward native wildlife. In many place around the world, other factors such as habitat loss, pollution, and climate change are seen as more urgent problems than introduced species.

Line 72-74: Not sure what this sentence means.

Line 90: A little awkward, what is the landscape of fear theorem? Never heard this phrasing before. Also, the landscape of fear refers to spatiotemporal variation in the risk of predation—this is distinct from “fear” (broadly defined) or stress. Can you be more clear about how this relates to the concept of stress as presented in this paper, both here and later (Lines 206-233).

Line 106: All manuscripts? Surely not—surely just all manuscripts that were included.

Line 135: Is there a reason why two stresses would be worse than one very intense stressor? If the authors are suggesting that stressors may be more than simply additive, they should consider explaining this in greater detail here.

Line 234: I disagree with this framing—a lot of behavioral ecology literature considers any cue stemming from a predator as a direct cue of risk, while the term indirect cue of risk is reserved for any situation that is associated with risk. Visual stimuli may also be important direct cues of risk, such as carcasses of conspecifics.

Line 253: It is incumbent upon the authors to explain here why olfactory cues of predators would not be directly correlated to risk of predation.

Line 283: “In general, human activity can be comparable to the impacts of predation”. What impacts? How is it comparable? Without more specific and precise writing, this section again is very hard to interpret.

Line 345: “…due to causes dissimilar to those driving global declines”. Not clear what this means.

Line 358: It’s not clear to me why this increased overlap between predators would lead to increased overall predation pressure on prey. Couldn’t you similarly argue that the outcome of this may be decreased predation, as the two predators might directly compete more strongly? Again, I think we need a little more context here.

Line 392: Why? Why is it insufficient to simply study demographic responses to foxes and land use change?

Line 400: This seems like a huge logical leap to me. Why would increased stress necessarily lead to a population collapse?

Line 438-447: I like the application of the landscape of fear here. I wonder if the authors could extend this application even further—is it enough to have low-stress refugia present throughout the landscape, or do we need these refugia to be functionally connected?

Line 480-484: I’m not sure these are all measures of stress. For example, giving up densities can provide information about, among other things, prey perception of risk (i.e., fear). But this strikes me as fundamentally different than stress, and so I encourage the authors to think critically about whether more detail/explanation needs to be included here.

Line 515: What are some examples of appropriate management?

References: Some citations are missing volume/page numbers (e.g., Gaynor et al. 2019). Please address this.

Reviewer 2 ·

Basic reporting

The authors present a thorough synthesis paper that aims to review the effects of predation stress and anthropogenic stress on wild animals, and argues for a consideration of these dual stressors in the conservation of animals living in human-impacted landscapes, using urban and suburban Australia as an example. To my knowledge, no review combining these aspects of predator-prey ecology, conservation, and physiology exists, and this paper is thus a useful addition to the literature. To make sure the paper can more effectively meet the stated goals, I would recommend some substantial revisions to bring out the main ideas of the manuscript and communicate them more clearly.

Experimental design

No comment

Validity of the findings

No comment

Additional comments

The authors present a thorough synthesis paper that aims to review the effects of predation stress and anthropogenic stress on wild animals, and argues for a consideration of these dual stressors in the conservation of animals living in human-impacted landscapes, using urban and suburban Australia as an example. To my knowledge, no review combining these aspects of predator-prey ecology, conservation, and physiology exists, and this paper is thus a useful addition to the literature. To make sure the paper can more effectively meet the stated goals, I would recommend some substantial revisions to bring out the main ideas of the manuscript and communicate them more clearly.

Generally, I noticed that important definitions are often introduced only after several sentences or paragraphs discussing a topic; I would recommend that the authors do a careful read-through and move definitions to the initial introduction of each key term to facilitate audience comprehension of the manuscript.

Additionally, the main idea of each section and paragraph was often buried in the middle of that section or paragraph, sandwiched between background information in a way that made it difficult to parse out what was background and what was a main point or new synthetic idea. The manuscript does have an important, unique perspective – that perspective would shine through more clearly with some succinct topic sentences at the beginning of each paragraph and section.

While I appreciate the thoroughness of the review, I think there was also occasionally more background information than was necessary to convey the main ideas, and some information presented was only tangentially relevant to the arguments of the manuscript. Some redundant background information could be cut, and this would also help improve the length of what is currently a long manuscript. To further clarify the aims and arguments of the paper, I would also recommend some editing of the organization of the manuscript and its sections. It could be useful to, as suggested in the previous comment, open each section with a guiding idea, provide background to contextualize this idea, and then return to the idea to flesh it out more thoroughly. Alternatively, most of the “review” portion of the paper – i.e. background on the different central concepts – could be at the beginning of the paper, and then have the second half of the manuscript focus on the authors’ arguments for how this information should be incorporated into conservation. This is how the paper is currently organized, to a certain extent, but more signposting of what is background/current knowledge and what is the authors’ new synthetic arguments would be ideal.

I also feel that missing from this manuscript is a discussion of the population-level consequences of stress and fear. There is literature that presents evidence for reductions in various measures of fitness (survival, fecundity, fat reserves, etc.) on a number of species in response to chronic stress, and this can have important implications for the survival of populations. Something to this effect would fit well in the first section of the main body or, perhaps, as its own section following the “Landscape of Fear” section. I would recommend increasing the attention paid to this new section and the discussion of conservation management strategies, and reducing the length of the section devoted to discussing detailed physiological responses to stress. Ultimately, I think this review has more value as an overview of the sources of stress that can impact vulnerable species (and the potential conservation actions that can be taken to alleviate this stress) than as a review of the physiology of stress responses in individuals.

Some more detailed comments are included below:

Beginning:

- I would recommend a few tweaks to the title. The word “stressing” sounds awkward; perhaps change it to just “stress”.
- I would also recommend some care in the abstract/author comment: the landscape of fear concept does not really support the manuscript’s contention so much as that contention is predicated upon the landscape of fear concept.

Introduction:

- I would suggest a slight restructuring of the introduction. Line 48 indicates that humans are an “additional” stressor to wildlife; however, no other stressors have as yet been identified at that point in the manuscript. As written, I think the introduction could set the reader up better if it opened with the overview of predator fear and stress impacts on prey, and then introduced the idea that the expansion of human activity can also present an additional stressor to wild animals.

Physiological Responses to Fear and Stress:

- I think it is important to specify early on in this section the types of organisms considered in this review; some sections of the discussion on physiology were pertinent only to animals with complex brains and central nervous systems, such as vertebrates, while others seemed to be more general. Is the manuscript meant to focus on vertebrates, or are all animal clades the subject of this review?

Behavioural Responses to Fear and Stress:

- The introduction to this section is fairly confusing as written. I would recommend introducing the landscape of fear concept (“concept” may be more appropriate than “theorem”), then clearly detailing how stress responses to predation in the landscape of fear can impact prey. In this section, Laundré et al. 2010 should also be cited, as they detail the impacts of stress responses in the landscape of fear more generally in that later paper. In the second paragraph (line 219), it is stated that “Theoretically, landscape of fear effects increase exponentially with increasing landscape homogeneity...(Bleicher 2018, Gaynor et al. 2019).” Technically, neither Bleicher nor Gaynor et al. make such an assertion, nor do I think it is true; Gaynor et al. state, and I agree, that in fact landscape of fear effects should increase with landscape heterogeneity, since the differences between high and low risk sites will be more pronounced and prey can more easily avoid high-risk sites. The sentences that follow are a better representation of the current thinking in the field, and I would restructure this section to emphasize these later points instead.

Human Activity as a Fear Inducing Stressor:

- The last paragraph of this section feels more like a topic (i.e. first) paragraph, but is also very long and I think too numerous in its examples. I’d recommend moving some of the information in this paragraph to the beginning of the section, but also greatly condensing it.

Management Tools to Observe and Alleviate Fear and Stress for Conservation:

- I like this section and I would like to see it expanded more relative to the review portion of the paper. It gets at the heart of the matter and contains some important ideas. I would recommend reflecting on the ideas presented in this section to help inform where to cut some of the background information earlier on in the paper. What information helps inform these recommendations, and what information has been presented in this manuscript that these new ideas and recommendations do not address? For example, as discussed above, I think the details of the physiological impacts of stress could be summarized more succinctly, but greater attention should be paid to the population- and species-level impacts of physiological stress responses, as these are the scales more relevant to conservation.

I liked the figures and thought they contributed nicely to the paper, particularly the checklist.

Line by line:

Abstract:

- I’d use “negative” or “are harming” instead of “deleterious” – may as well make the language as clear as possible.
- “Conservation management is now beginning to consider the effects of human impacts as tantamount to or surpassing the impacts of introduced predators”: I would recommend toning down this language, as I would argue that management is not only just now beginning to consider human impacts on wildlife, but has been grappling with this for some time; indeed, this is central to the field of conservation. It is just as effective to say that simultaneously [in reference to the prior sentence about introduced predators] human populations also have significant impacts on wild animals.
- I would remove the sentence that presents Australia as an example – while you do extend this example in the manuscript, I don’t think it is necessary in the Abstract and it is distracting as a threat that is picked up and rapidly put down.
- The authors don’t “aim to highlight the need for future studies” – they DO highlight this need! They can remove the words “aim to” as they have succeeded in this goal.
- I found the last sentence of the abstract somewhat confusing, and particularly wasn’t sure what the clause “along with habitat influences” was referring to – I’d recommend clarifying and simplifying this sentence.

Main Body:

- Line 55: The use of “while” suggests that the following clause is in contrast with the previous statement. However, eliciting fear responses is itself one of the ways that predators influence ecosystem functioning (see Schmitz et al. 2004, Schmitz et al. 1997, Hawlena and Schmitz 2012)
- Line 58: This is a slightly misleading/vague definition of the “landscape of fear”. This is an important framing concept for the paper, so I would be more specific and careful with this definition. The landscape of fear is not so much the costs and benefits that prey balance, but the spatial distribution of perceived predation risk that influences prey movement and behavior as they attempt to mitigate risk and obtain essential resources.
- Line 70: But see Zbyryt et al. 2017
- Line 178: This sentence should be towards the beginning of the paragraph, before you begin to discuss fear responses.
- Line 207: I don’t really understand this sentence – it could be rewritten to be more clear.
- Line 259: This isn’t quite the definition of the predation-sensitive food hypothesis as outlined by Sinclair and Arcese; rather, the PSFH states that prey should take more risks when food is scarce and their need for food outweighs their fear of predation. Be careful with definitions here and make sure the papers cited truly support each statement.
- Line 262: Also Smith et al. 2019
- Line 300: Should also cite Gaynor et al. 2018 here
- Line 302: See also Kohl et al. 2018, Smith et al. 2019
- Line 432: This is a lot of references – I think you only need a few examples to bolster this statement.
- Line 482: There is some question as to whether giving-up densities accurately measure predation stress, given the impacts of stress on physiology and thus dietary preference. See McMahon et al. 2018, should probably cite this paper as an alternative viewpoint.

The following papers are also important literature on the topic and should be engaged with in the manuscript (some of these have been mentioned explicitly in the line-by-line comments):

Smith JA, Donadio E, Pauli JN, Sheriff MJ, Middleton AD. 2019. Integrating temporal refugia into landscapes of fear: prey exploit predator downtimes to forage in risky places. Oecologia 189:883–890. DOI: 10.1007/s00442-019-04381-5.
Smith JA, Suraci JP, Clinchy M, Crawford A, Roberts D, Zanette LY, Wilmers CC. 2017. Fear of the human “super predator” reduces feeding time in large carnivores. Proceedings of the Royal Society B: Biological Sciences 284:20170433. DOI: 10.1098/rspb.2017.0433.
Suraci JP, Clinchy M, Zanette LY, Wilmers CC. 2019. Fear of humans as apex predators has landscape-scale impacts from mountain lions to mice. Ecology Letters 0. DOI: 10.1111/ele.13344.
Clinchy M, Sheriff MJ, Zanette LY. 2012. Predator-induced stress and the ecology of fear. Functional Ecology 27:56–65. DOI: 10.1111/1365-2435.12007.
Say-Sallaz E, Chamaillé-Jammes S, Fritz H, Valeix M. 2019. Non-consumptive effects of predation in large terrestrial mammals: Mapping our knowledge and revealing the tip of the iceberg. Biological Conservation 235:36–52. DOI: 10.1016/j.biocon.2019.03.044.
Vijayakrishnan S, Kumar MA, Umapathy G, Kumar V, Sinha A. 2018. Physiological stress responses in wild Asian elephants Elephas maximus in a human-dominated landscape in the Western Ghats, southern India. General and Comparative Endocrinology. DOI: 10.1016/j.ygcen.2018.05.009.
Zbyryt A, Bubnicki JW, Kuijper DPJ, Dehnhard M, Churski M, Schmidt K, Wong B. Do wild ungulates experience higher stress with humans than with large carnivores? Behavioral Ecology. DOI: 10.1093/beheco/arx142.
Kohl MT, Stahler DR, Metz MC, Forester JD, Kauffman MJ, Varley N, White PJ, Smith DW, MacNulty DR. 2018. Diel predator activity drives a dynamic landscape of fear. Ecological Monographs 88:638–652. DOI: 10.1002/ecm.1313.
Gaynor KM, Hojnowski CE, Carter NH, Brashares JS. 2018. The influence of human disturbance on wildlife nocturnality. Science 360:1232–1235. DOI: 10.1126/science.aar7121.
Schmitz OJ, Beckerman AP, O’Brien KM. 1997. Behaviorally mediated trophic cascades: effects of predation risk on food web interactions. Ecology 78:1388–1399.
Hawlena D, Schmitz OJ. 2010. Physiological stress as a fundamental mechanism linking predation to ecosystem functioning. The American Naturalist 176:537–556. DOI: 10.1086/656495.
Laundré JW, Hernández L, Ripple WJ. 2010. The landscape of fear: ecological implications of being afraid. Open Ecology Journal 3:1–7.

·

Basic reporting

There is a sizeable amount of background information and literature review in this manuscript, and the review is certainly of broad and cross-disciplinary interest. The combination of themes – from the landscape of fear theorem, to stress physiology, and conservation biology – are themes that are certainly applicable and timely. There have been recent reviews on several of the topics individually (Gaynor et al. 2019 in TREE reviews the landscape of fear; McCormick & Romero 2017 review conservation endocrinology in BioScience), but there are not many reviews that integrate the cadre of perspectives into a cohesive narrative. This manuscript begins to broach that cross-disciplinary divide

The introduction does a good job of introducing the subject for sure; however, the scope of the manuscript is not entirely clear or consistent throughout the manuscript. There is mention that this is helpful for wildlife conservation in the final paragraph of the manuscript, but a good chunk of the latter half of the manuscript talks exclusively about prey species, particularly small mammals. I actually think that leaning into small mammal conservation is a more sufficient move the way that the manuscript is written, which many of the subsections that talk about Australia-specific conservation issues set the stage for later exploration of how stress physiology and endocrine studies can help in that regard.

In sum, changing the focus of the intro a bit to shift more to prey species conservation, and potentially the residual, bottom-up effects of doing so, would make the argument of the manuscript clearer.

Experimental design

The survey methodology is consistent and comprehensive, certainly no issues there. In fact, part of the potential pitfall of the manuscript is that the comprehensive review contained too many references, without distilling the relevant information to the argument at hand. There could feasibly be some reorganization of the text to highlight what I gathered to be a logical argument: (1) Australian wildlife – especially small mammals – are being threatened by habitat fragmentation and introduction of invasive predators; (2) Stress physiology and fear dynamics could be used as a tool to create solutions that mitigate loss of threatened or endangered species; (3) urban planning in combination with ecophysiology and behavior could be informed greatly by implementing endocrine function and behavior into habitat construction. At the moment, there is a heavy focus on describing the worlds of stress physiology and the landscape of fear theorem that the ultimate goal of the manuscript gets lost in the minutiae.

Validity of the findings

Though the information in this review is great and substantial, it is hard to follow along and assess what the ultimate goals are of the manuscript. The section on stress physiology defines a lot of the terms and background behind endocrine function, but should focus more on describing the trends that have been previously found in the literature. Work by Francis Bonier and others have done an incredible job demonstrating how stress physiology in birds is all over the map: some species show increases in baseline CORT, some demonstrate decreases, and others no difference as a function of urbanization; which, can serve as a proxy for human activity. The nuance and complexity of stress responses toward human beings is not made clear or elaborated and delving more into the literature about the connections among human activity and endocrine responses would be helpful. I might suggest putting together a summary table that has a species for each row and the direction of stress response toward a stressor or in alignment with a proxy measure for human activity. Bonier 2012 in Hormones and Behavior is a good start.

For the landscape of fear section, making sure to hone in on a specific trophic level (again, I think the most compelling narrative would be around small mammals) is important, as it seems as though there is a lot of definitions of terms, but not a true synthesis of the material. In some instances, there seems to be a bit of redundancy, which could be solved by focusing on small mammal responses to predators and humans. Again, there is a rich literature in fear responses and habituation in species that spend significant amounts of time with humans, including work by Honda et al. 2018 in Science of the Total Environment.

Shoring some of these points up, then making explicit how incorporating stress physiology and human activity helps to create specific strategies that aid in improving conservation, will help to bring the point home a bit more. In that same regard, being more specific about what human activities exacerbate fear dynamics, and which ones actually lead to human shields, is an important distinction that needs to be made in order for the manuscript to bring things home. Because human activity does not always result in fear responses, it is reasonable to predict that human activity may not always elicit acute or chronic stress responses, which would suggest that human activity in some instances is actually a good thing. Paired with the fact that the human shield hypothesis would suggest that prey species are more protected from natural predators, then it may be possible that small mammals in urban or suburban regions see fitness benefits (albeit, potential costs may include dealing with cats, but this narrative is not fleshed out in the current draft). Spending more time talking about these complex differences and disaggregating human activity to talk about the types of human activities and how they generate differences across the landscape would potentially be powerful.

---

## Round 0.2 · Minor Revisions

The reviewers differed quite a bit in their response to your revised manuscript which perhaps reflects the challenges of integrative reviews across what are typically distinct sets of literature. I agree with various perspectives of each reviewer depending on the section of the manuscript which makes a decision on this manuscript challenging.

Overall I relate strongly with all reicewers' sentiments that this manuscript does a thorough job of reviewing and summarizing the state-of-knowledge in the fear and stress literatures. However, I also agree with Reviewer 1's comments that this review doesn't hit the mark on providing an integrative framework for how these fit together cohesively and in an actionable way. One thing I walk away with is a big question as to whether stress and fear are additive or interactive effects, how we assess this, and how management chips away at either problem independently or in tandem. Your methods section claims your manuscript is a "novel synthesis". We need to see a synthesis

I stress this in relation to the criteria for review papers (https://peerj.com/about/editorial-criteria/#literature-review-criteria). The one sticking point is the validity of findings with respect to whether the review meets the goal that it has laid out. In this case, the manuscript is a great review of a couple fields but the integration as a synthesis doesn't seem to quite get there. Of course, there is subjectivity in this criterion for review paper.

One way to make this review more synthetic may be to develop a conceptual framework figure that illustrates the various direct and indirect ways urban human activities as well as predators influence wildlife stress and then highlight areas where management targets may intervene to mitigate stress. Something along the lines of a flow diagram or network / web could be informative. Adding a short section / paragraph after the “Human activity…” section that mirrors the connections in this figure and which discusses the interconnectedness of the various themes reviewed I think would go a long way to turning this manuscript more into a synthesis that has a tractable message rather than a review of several disjunct fields. This exercise might also help the authors find more ground to provide more detailed management recommendations as Reviewer 1 indicated – and I agree – the management suggestions are a little sparse leaving little to walk away from with regards to actionable items. I think these additions are relatively minor but would go a long way to improving the usefulness of this review.

Reviewer 2 also makes an interesting point of bringing the idea of supplementation into the main text. I think this is important and encourage you to make sure the various points in your figures are meaningfully reflected in the text.

With respect to reviewer 1’s comment about accurate language, please modify the sentence surrounding the Zbyryt et al., 2017 citation accordingly and take care throughout your manuscript to sure your review and synthesis accurately portrays the referenced studies.

In your abstract and main text, please rephrase urban areas as “hotspots” of wildlife activity to something more realistic.

Line 388: you incorrectly use the word “adaptation” here as an umbrella term for both plastic and evolved capacities of organisms to deal with urban environments. Please modify to something like “The ability of wildlife to cope with urban environments can occur through either plastic or evolved shifts in behavior, foraging…”. In general this paragraph is problematic because it suggests that adaptive evolution to cities is more common than plastic responses to urbanization. While there has been much discussion in recent years about rapid adaptation to cities, the evidence is still very faint and plasticity still remains the most likely cause of species coping with urban stress. I’d recommend reframing this section to make it clear that most species likely plastically adjust to urban stress with some of them actually ultimately adaptively evolving tolerance to urban environments. This would be a more accurate representation of the state of the literature.

Finally, Reviewer 2 highlights a number of line-by-line comments. The manuscript is a bit "clunky" to read. I strongly encourage you to take care with wording and grammar issues as well as the readability of your manuscript before resubmitting it.

Depending on the nature of the revision, I may send the manuscript back for review. I do look forward to seeing a revised version of this manuscript.

·

Basic reporting

Fardell et al. review how predators affect both the ecology and physiology of their prey, and argue that conservation efforts (particularly of mammals, for which we have a rich ecological and physiological literature) could be improved by accounting for physiological stress in conservation. I reviewed an earlier draft of this manuscript, and I appreciate the work the authors have done in addressing all three original reviews. In particular, the authors have shored up the literature cited, revised the organization of the paper, and clarified text throughout the paper.

Unfortunately, the revised manuscript treats predator-prey ecology and physiological/neurological stress quite separately throughout the paper, failing to provide a direct, integrative link between predator-prey ecology and stress. I raised this concern in my first review, and the authors state in their response letter that they did not feel it would strengthen their argument. I strongly disagree, as a true cross-disciplinary synthesis needs to show exactly how each discipline can be applied to the other, rather than simply providing a side-by-side review of literature in each discipline. I will provide an example to demonstrate my point. In the section titled “Managing habitat to alleviate wildlife fear and stress”, the authors write the following paragraphs (I’m paraphrasing the topic sentences here): Paragraph A: the importance of configuration of predation refuge; Paragraph B: the importance of structurally complex habitat; Paragraph C: animals prefer to forage in safe places/times; Paragraph D: humans reduce habitat complexity. In this whole section, I could only find two sentences in Paragraph C that explicitly mention the role of stress (Line 494-496), and even here it is not clear what managers can actually do to reduce the stress felt by animals. I’d like to point out that I don’t think the authors have done a poor job of reviewing the predator-prey literature in this paper—there is a lot of good review work here and I think the authors have done a good job tying together literature about refuge, foraging ecology, and the landscape of fear. However, in my opinion, the review falls short of its stated goal of synthesizing stress and fear literature, which is unfortunate because I think these disciplines do have a lot to learn from each other.

Line 83: I find this wording to be very misleading. A quick look at the Zbyryt et al. (2017) article shows that Zbyryt et al. compared areas of high and low human activity or predator activity, and observed that the lowest stress levels occurred in areas with high predator/low human activity, which prompts Zbyryt et al. to conclude that prey exposed to constant risk may have adapted different responses to stress. I realize that this may seem extremely picky, but the authors’ wording here misrepresents the study’s experimental design (“increase” and “decrease” suggest some experimental manipulation and tracking subjects over time) and thus, intentionally or not, causes readers to misinterpret the original study. I find this particularly problematic for a synthetic review paper, which is tasked with synthesizing and summarizing existing literature—the use of imprecise language in a review paper can in fact potentially generate more confusion. I would suggest that the authors carefully screen the paper for any other potential instances of misleading wording.

Experimental design

No comment

Validity of the findings

Throughout the paper, the authors present stress and fear as something that humans should seek to minimize in prey. For example, Line 511 is a subtitle “Management tools to alleviate fear and stress for wildlife conservation”. Similar language is found in the first sentence of the Abstract, pointing out the negative effects of predators on biodiversity, but not their positive effects. (Line 25). In my experience, this is not always the case—predators provide many important services to society by controlling their prey (Estes et al. 2011 Science provides a great starting point). Taking such value-laden wording out of this paper would help it provide a more objective tone, which would consequently help its application to management (as predators may have net positive effects in some cases and net negative effects in others).

I’m not sure that the authors present a clear plan to help managers assess fear and stress (Line 511-534). The authors state, “Stress in wildlife may be cost-effectively and straightforwardly observed by combining the outcomes of simultaneous measurements from multiple well-established methods” and then go on to mention giving-up-densities, glucocorticoid assays, and habitat assessments. How are managers to make sense of and integrate these three very different types of data? Giving-up-densities—which I am most familiar with here—are difficult and time consuming to pull off, and not without their pitfalls. Moreover, they simply measure the perceived cost of foraging; there’s more nuance in using them to assess fear than the authors present here (how can metabolic costs and missed opportunity costs realistically controlled for?). The authors go on to provide their “reactive scope model” based CEA (Line 545-561), but it is not clear to me how the methods described above (giving-up-densities, glucocorticoid assays, and habitat assessments) could be used to inform or parameterize such a model, or how to interpret the output of this model.

Additional comments

I appreciate the clear definitions of fear and stress added by the authors (Lines 143-167). However, I think that adding a brief (as in, a few words) definition on Line 50 would help even further, since the concept of fear is introduced here before the reader gets a chance to digest the more comprehensive definitions of stress/fear starting on Line 143.

Line 31: Urban areas as “global hotspots of wildlife activity” seems a bit of a stretch. I would recommend rephrasing this to something like, “important habitat for some wildlife species”.

Line 32: Do we need a brief clarification of the operational definition of “fear responses” here, too?

Line 184: This sentence was a bit unclear to me. It seems like the authors are pointing this out as a drawback—why exactly is it a drawback?

Line 317: How does food limitation interact with risk to affect fear/stress in cities?

Line 349: Unclear to me what a “hunting impact” is. Do you mean hunting intensity?

Line 452: What would managing vegetation structure look like?

Line 475: An interesting counterpoint to this is that in eastern North America, invasive shrubs provide very safe habitat for small mammals (Mattos and Orrock 2010 Behavioral Ecology ), but have low biodiversity.

Line 513: How does simply indicating an area as needing intervention provide returns on restoration investment?

Line 522: What is meant by “spatially correlating mapping resources”?

Line 566: In what ways are small mammals declining in Australia? Population sizes, taxonomic diversity? Both?

Reviewer 2 ·

Basic reporting

Overall, I am very happy with this revision – the revised manuscript is much clearer and easier to read than the prior draft, and provides an important contribution to the literature by simultaneously reviewing the stress impacts of predation and human activity on mammals. The authors close by providing recommendations for conservation management that take into account stress impacts from native predators, introduced predators, and human activity. I have only minor comments remaining, and very much enjoyed read this thorough and constructive review.

Experimental design

This is a well-designed literature review.

Validity of the findings

The authors satisfactorily review the literature on predator- and human-induced stress responses in mammals and provide sound conservation recommendations.

Additional comments

My only remaining general comment relates to Figure 2. In the figure, and associated caption, the authors recommend supplementation of resources such as water or nest boxes as an additional conservation measure in high-stress environments – however, this recommendation does not feature prominently in the main text. I think it is an interesting idea, since stress can inhibit an animal’s ability to obtain vital resources while minimizing risk, though it could also have unintended consequences (e.g. bolstering populations of introduced predators and competitors, providing high-use sites where predators can easily pick off prey). I think that if the recommendation is to remain in Figure 2, which I think it should, it should also be discussed more extensively in the text, in the same section where engineering habitat structure in discussed as a conservation measure for stress reduction. I understand that the paper is already long; however, I think just a few sentences or a short paragraph would suffice to discuss the suggested management approach, its potential benefits, and any potential negative consequences to be considered.

Line-by-line Comments:

- The sentence in lines 62-64 is quite awkard/confusing; would reword in a more active voice “Climate change and associated shifts in primary productivity can also have bottom-up impacts on predator-prey interactions.”
- Line 87 – would change “discipline” to “disciplinary”
- Line 103 – would remove “ – in the context of the landscape of fear”, I think it makes the sentence a little long and clunky and the point can be left for the main body.
- Line 340 – would say “than in top predators”
- Line 343 – remove comma after “above-mentioned”, replace em-dash and “including” with comma
- Line 344 – change “, and perhaps even stronger if it adversely disrupts foraging” to “and disrupting foraging”
- Line 345 – Start new sentence: “Humans can also act as super-predators ...”
- Line 347 – would reword sentence to “Trophic cascades can also arise when human activity has non-consumptive effects on the ecological roles of large predators”
- Line 356 – would refer to “top-down” and “bottom-up” effects with the hyphen throughout
- Line 372 – change “traffic/roads/vehicles” to “roads and vehicles”
- Line 372 – remove “such as snow sports/marine/sports/land sports” – recreational activities is clear enough and having more words complicates the sentence
- Line 373 – replace “otherwise” with “designated”
- Line 373 – replace “can manifest as physiological stress impacts or in the complete displacement of wildlife, particularly predator species, and outweigh any positive effects” with “can lead to physiological stress or displacement, countervailing positive effects of humans on wildlife.” – reads as confusing as is.
- Line 411 – replace “; namely:” with “, including”
- Line 438 – I would delete “via cumulative impacts....naturally in ecosystems”; let the figure explain!
- Line 470 – I think this is far more citations than is needed to justify this point; this occurs in a few other places as well. If the journal prefers that the reference list be cut, and also to ease the flow of reading (which can be hampered by interruptions of long reference lists), the authors could choose a review paper and a few seminal papers to make this and other points. I defer to the editor, but generally five citations is more than enough to bolster a general point such as this.
- Line 492 – change “in landscape of fear habitats” to “in landscapes of fear,”
- Line 500 – delete “and thus should be of particular importance in such management in the future” – this point is made later in the paragraph and it makes the sentence longer and harder to read than necessary
- Line 569 – change these semi-colons to commas
- Line 585 – cite this study in this sentence, not the following one.

Figure 1: In the caption, perhaps the authors can explicitly list the introduced stressors represented in the figure, as is done for the habitat stressors.

·

Basic reporting

The authors have incorporated a substantial amount of edits to clarify many of the important points of their review and highlight the compounding influences of predators and urbanization on prey species.

Experimental design

The study design has been further clarified for the reader and the reorganization of text has helped in that regard as well. Well done!

Validity of the findings

I very much appreciate the increased emphasis on application of stress physiology to increasing conservation and management efforts, and think that this was done particularly well.

Additional comments

Great work!

---

## Round 0.3 · Minor Revisions

Your manuscript has now been assessed by myself and a reviewer who has seen all versions of this paper. We both agree this manuscript is greatly improved and better matches its goals of synthesizing concepts from predator ecology, human-wildlife conflict, and stress biology into conservation. I appreciate your thorough responses to the reviewers and revision of this manuscript.

The reviewers has only a small number of minor edits and I'll emphasize their points about editing lines 25 and 419-433. Also, while this certainly isn't mandatory, I still believe a conceptual figure would greatly improve the easy with which this manuscript could be digested and used.

I believe these small number of edits should be straightforward and will not require further review. I look forward to receiving your revision.

Reviewer 2 ·

Basic reporting

I think this review has improved greatly over previous drafts – in particular, the language and organizational structure is much more clear. I am also pleased with the addition of the discussion of the pros and cons of supplementation, and would now recommend the paper for publication. I have included a few small language edits below that could be addressed in minor revisions or at the proof stage, if the authors and editor see fit.

Experimental design

No comment

Validity of the findings

No comment

Additional comments

Line-by-line:

Line 25: This first sentence I think is a little misleading, in that it frames all predators in a negative light despite the fact that native predators are important components of ecosystems. Perhaps change to something that indicates that predators induce stress effects in prey generally [less negative, just a statement of fact], and introduced predators can have particularly harmful effects on native biodiversity. This is similar to what the authors do well in the introduction.
Line 71: I would replace “that is” with a comma, reads more clearly that way.
Line 87: Replace “it” with “stress”
Line 240: Perhaps leave the animal vulnerable to future “threats” rather than “stressors”? Since part of the point is that the animal would not exhibit a stress response to something it should – so that threat is not a stressor, in the physiological sense, but still presents a tangible threat to the animal’s fitness
Line 248: Replace “show variable reactions due to” with “show that reactions vary with”
Line 254: Change to “a consistent stress response pattern has yet to be observed in birds”
Line 343: Change “among” to “within”
Line 357: I would change “top-down” to “trophic”; trophic cascades are inherently top-down
Line 404: Remove both commas
Line 405: “Behavior, endocrine, and other traits” is awkward wording; do you mean to refer to these as nouns or adjectives?
Line 411: Switch order to “increasingly observed”
Line 421: Remove semicolon
Line 419-433: I don’t think this paragraph is necessary. It’s a grab bag of all the ways we know humans negatively impact biodiversity, which has been well reviewed in other papers and is ultimately not what this paper is about, and it doesn’t include any analysis beyond listing these impacts. This paper specifically deals with stress impacts on animals living near humans, which are thoroughly covered in the rest of this section. I think it could be easily cut.
Line 492: Would reword to “include increased parasite resistance in nestling birds after supplementation with high-quality food during the stressful young-rearing stage”. Then separate this really long sentence with two examples into two sentences. The second clause (about corridors) is very confusingly worded, I’m not really sure what it’s trying to say. Perhaps reword in active voice that clearly indicates a) the problem, b) the intervention that was made and C) the positive result that occurred – it seems to currently be telling that story backwards.
Line 620: Change “was” to “were”

---

## Round 0.4 · accepted · Accept

Thank you for your continued efforts in revising this manuscript in accordance with the reviewers' suggestions. Congratulations on your synthesis paper!